# Recovery of Recombinant Avian Paramyxovirus Type-3 Strain Wisconsin by Reverse Genetics and Its Evaluation as a Vaccine Vector for Chickens

**DOI:** 10.3390/v13020316

**Published:** 2021-02-19

**Authors:** Mohamed A. Elbehairy, Sunil K. Khattar, Siba K. Samal

**Affiliations:** 1Virginia-Maryland Regional College of Veterinary Medicine, University of Maryland, College Park, MD 20740, USA; melbehai@umd.edu (M.A.E.); sunilk8@gmail.com (S.K.K.); 2Department of Poultry Diseases, Faculty of Veterinary Medicine, Cairo University, Giza 12211, Egypt

**Keywords:** avian paramyxovirus type 3 (APMV-3), paramyxovirus vaccine vector, reverse genetics, virus pathogenesis, avian vaccine

## Abstract

A reverse genetic system for avian paramyxovirus type-3 (APMV-3) strain Wisconsin was created and the infectious virus was recovered from a plasmid-based viral antigenomic cDNA. Green fluorescent protein (GFP) gene was cloned into the recombinant APMV-3 genome as a foreign gene. Stable expression of GFP by the recovered virus was confirmed for at least 10 consecutive passages. APMV-3 strain Wisconsin was evaluated against APMV-3 strain Netherlands and APMV-1 strain LaSota as a vaccine vector. The three viral vectors expressing GFP as a foreign protein were compared for level of GFP expression level, growth rate in chicken embryo fibroblast (DF-1) cells, and tissue distribution and immunogenicity in specific pathogen-free (SPF) day-old chickens. APMV-3 strain Netherlands showed highest growth rate and GFP expression level among the three APMV vectors in vitro. APMV-3 strain Wisconsin and APMV-1 strain LaSota vectors were mainly confined to the trachea after vaccination of day-old SPF chickens without any observable pathogenicity, whereas APMV-3 strain Netherlands showed wide tissue distribution in different body organs (brain, lungs, trachea, and spleen) with mild observable pathogenicity. In terms of immunogenicity, both APMV-3 strain-vaccinated groups showed HI titers two to three fold higher than that induced by APMV-1 strain LaSota vaccinated group. This study offers a novel paramyxovirus vector (APMV-3 strain Wisconsin) which can be used safely for vaccination of young chickens as an alternative for APMV-1 strain LaSota vector.

## 1. Introduction

The family *Paramyxoviridae* contains pleomorphic, enveloped viruses with a non-segmented, negative-sense RNA genome. Members of this family have been isolated from a wide variety of avian and mammalian species around the world, which includes many important human, animal and avian pathogens [1]. The family *Paramyxoviridae* is divided into four subfamilies; *Avulavirinae, Orthoparamyxovirinae, Metaparamyxovirinae* and *Rubulavirinae* [2]. All avian paramyxoviruses (APMVs) are placed under the subfamily *Avulavirinae* in three genera: *Orthoavulavirus, Metaavulavirus,* and *Paraavulavirus*. Currently, there are twenty officially recognized APMV species. In the latest International Committee on Taxonomy of Viruses (ICTV) classification APMV-1 was placed in genus *Orthoavulavirus,* while APMV-3 was placed in genus *Metaavulavirus* [2,3].

APMV-1 is the best characterized member among APMVs because its virulent strains, known as Newcastle disease virus (NDV), cause a highly contagious disease with major economic importance in chickens worldwide [4]. However, our knowledge about replication and pathogenicity of other APMVs is very limited. The complete genome sequences of one or more representative strains of other APMVs have been reported [3]. The genome lengths of all APMVs range from 15 to 17 kb. Most APMV genomes consist of six genes: N (nucleocapsid), P (phosphoprotein), M (matrix protein), F (fusion protein), HN (hemagglutinin-neuraminidase protein), and L (large polymerase protein); except APMV-6 which has an additional SH (small hydrophobic) gene [3]. 

To date, reverse genetics systems have been developed for APMV-1 [5], APMV-2 [6], APMV-3 [7], APMV-6 [8], APMV-7 [9], and APMV-10 [8]. The reverse genetics system of APMV-1 has greatly benefited our understanding of its replication and pathogenesis [4]. In addition, it has been used as a vaccine vector for animal and human pathogens [10,11]. However, the potential of reverse genetics systems of other APMVs has not been fully evaluated. 

The disease potential of APMV-1 has been well studied [4]. APMV-2, APMV-3, APMV-6 and APMV-7 have been associated with mild respiratory disease in poultry [3]. APMV-3, was first isolated from turkeys with respiratory tract disease in Ontario, Canada, in 1967 and then in Wisconsin, USA, in 1968 [12]. Since then, APMV-3 strains have been isolated from turkeys in England, France and Germany [1]. However, most APMV-3 isolations have been from psittacine and passerine birds held in quarantine [13,14]. There are two distinct strains of APMV-3 with varying pathogenicity in chickens. APMV-3 strain Netherlands is mildly pathogenic to young chickens, whereas APMV-3 strain Wisconsin is non-pathogenic to young chickens [15,16,17,18]. 

The complete genome sequences have been determined for APMV-3 strain Netherlands and APMV-3 strain Wisconsin [17,18]. Both strains share 67% nucleotide identity and 78% amino acid identity. Antigenic analysis by cross-HI and cross-neutralization tests showed that both strains belong to the same serotype but represent two antigenic subgroups [17]. The F protein cleavage site of APMV-3 strain Netherlands has a multi-basic amino acid motif, similar to that of virulent APMV-1 (NDV) strains, whereas APMV-3 strain Wisconsin has a monobasic amino acid motif at its F protein cleavage site, similar to that of avirulent APMV-1 strains (Appendix A) [17,18,19]. 

A reverse genetics system has been developed for APMV-3 strain Netherlands and the recombinant virus has been used as a vaccine vector to evaluate the role of NDV F and HN proteins in the protective immunity [7]. Recently, APMV-3 strain Netherlands was used successfully as a vaccine vector for protection of chickens against HPAI (H5N1) [20]. It was also found that the P-M gene junction is the optimal insertion site in the genome of APMV-3 strain Netherlands for foreign gene expression [21]. APMV-3 strain Netherlands expressing Ebola virus glycoprotein was found to elicit mucosal and humoral immune responses against the Ebola virus glycoprotein in guinea pigs [22]. These results indicate that the recombinant APMV-3 strain Netherlands has great potential as a vaccine vector for veterinary and human uses. One advantage APMV-3 has over APMV-1 as a poultry vaccine vector is that it shows minimal cross reactivity with the maternal antibodies present in commercial chickens against NDV. This advantage prevents neutralization of the vaccine vector when used in chickens with maternal antibodies NDV [7,20]. In addition, APMV-3 strains are frequently isolated from turkeys; therefore, they may be successful vaccine vectors for turkey vaccination.

APMV-3 strain Netherlands is considered the prototype of APMV-3 [15], and it has some pathogenic effects in day-old chickens [23]. It causes stunted growth (especially in young broiler chickens) and kills the embryo during propagation in embryonated chicken eggs (ECE) [16,23]. By contrast, APMV-3 strain Wisconsin was reported to be nonpathogenic to young chickens and does not kill chicken embryos during propagation in ECE [16]. Therefore, we hypothesized that APMV-3 strain Wisconsin may offer a safer vaccine vector for day-old chickens or in ovo vaccination. To test this hypothesis, we developed a reverse genetics system for APMV-3 strain Wisconsin and recovered the recombinant virus. The recombinant APMV-3 strain Wisconsin was tested to express the green fluorescent protein (GFP) as a foreign protein and compared with recombinant APMV-3 strain Netherlands and APMV-1 strain LaSota vectors expressing GFP in vitro and in vivo.

## 2. Materials and Methods

### 2.1. Viruses, Cells and Animals

Chicken embryo fibroblast (DF-1) and human epidermoid carcinoma cells type-2 (HEp-2) were grown in Dulbecco’s minimal essential medium (DMEM) supplemented with 10% fetal bovine serum (FBS). Cells infected by APMVs were maintained in DMEM supplemented with 2% FBS and 10% chicken egg allantoic fluid (as a source of exogenous protease). The viruses used in this study were; APMV-3 strain turkey/Wisconsin/68 (APMV-3 Wisc.), obtained from national veterinary services laboratory, Ames, Iowa. Recombinant APMV-1 strain LaSota (rLaSota) and recombinant APMV-3 strain parakeet/Netherlands/449/75 (rAPMV-3 Neth.) expressing GFP were prepared in our laboratory with the GFP gene cloned into their P-M gene junctions. Modified vaccinia virus strain Ankara expressing T7 polymerase (MVA-T7) was obtained from Dr. Bernard Moss (National Institute of Allergy and Infectious Diseases). The avian paramyxoviruses were grown in 10-day-old specific pathogen-free (SPF) ECE by intra allantoic inoculation. The SPF chickens and ECE were obtained from Charles River Laboratories, Manassas, VA, USA.

### 2.2. Construction of Avian Paramyxovirus 3 Strain Wisconsin (APMV-3 Wisc.) Antigenomic Full-Length Plasmid

APMV-3 Wisc. RNA was isolated from purified virus using Trizol reagent (Invitrogen™) following the manufacturer’s protocol. A complete virus antigenomic cDNA was created using the viral genomic RNA, superscript reverse transcriptase IV (Invitrogen™) and short random primers (hexamers) following the manufacturer’s protocol. The viral genome was divided into six major fragments (I-VI) (corresponding to the six virus genes) using restriction enzyme (RE) sites. All RE sites were introduced in the downstream untranslated region of each gene except for the L-gene where three RE sites, naturally present in its open reading frame (ORF), were used to sub-divide the L-gene into four smaller fragments (Figure 1A). Three RE sites, two RsrII and one SacII, in the ORF of N, P, and F genes, respectively, were deleted by silent mutagenesis, thus they can be used for other genome fragments cloning. For that reason, fragments I, II and IV were synthesized in two pieces and connected by overlapping polymerase chain reaction (PCR) to delete RsrII and SacII sites. Plasmid pBR322/dr was used as a backbone vector to clone the virus full length antigenome. Plasmid pBR322/dr was previously prepared by modifing of low-copy-number plasmid pBR322 to include the T7 RNA polymerase promoter, a polylinker and the hepatitis delta virus (HDV) ribozyme sequence [24]. 

A multiple cloning site oligonucleotide (polylinker) was designed to contain RE sites used for cloning of the complete virus genome (Appendix A). The polylinker was cloned into the pBR322/dr plasmid between AscI and RsrII sites. This cDNA was used as a template to create the virus subgenomic fragments by PCR using primers bearing the RE sites (Appendix A) and high-fidelity platinum pfx polymerase enzyme (Invitrogen™). The subgenomic fragments were cloned sequentially into the prepared vector. The L gene was cloned first in a reverse order (Figure 1A; starting from VIa to VId) followed by insertion of other fragments to form the antigenomic full-length clone (FLC) of APM3 strain Wisconsin. Each fragment was confirmed for absence of any unintended mutation by sequence analysis using a big dye terminator kit. A T7 RNA polymerase promotor sequence was inserted before the virus anti-genomic leader and a the HDV ribozyme sequence was inserted after the virus anti-genomic trailer. The resulting APMV-3 Wisc. full-length expression plasmid was termed “pAPMV3 Wisc. FLC.” (Figure 1A).

### 2.3. Construction of APMV-3 Wisc. Support Plasmids

Reverse transcription PCR (RT-PCR) was used to create APMV-3 Wisc. N and P genes ORF cDNA, which was cloned into the expression plasmid pTM (pTM-N and pTM-P) (Figure 1B). RE used for cloning the N gene were NcoI and SpeI, and for the P gene were NcoI and XhoI. APMV-3 Neth. L gene was used instead of APMV-3 Wisc. L gene for its recovery. APMV-3 Neth. L gene cDNA was previously cloned into the expression plasmid PcDNA3.1 using XbaI and NheI RE [7] (Figure 1B). Both pTM and PcDNA3.1 plasmids had T7 polymerase promotor sequence. The cloned genes were sequence confirmed before being used in virus recovery. 

### 2.4. Recovery of Recombinant APMV-3 Wisc.

Recovery of infectious recombinant APMV-3 Wisc. (rAPMV-3 Wisc.) was carried out using the constructed plasmids following our established protocol [5]. Briefly, HEp2 cells were co-transfected with the plasmids pAPMV3 Wisc. FLC (5 µg), pTM-N (3 µg), pTM-P (2 µg) and PcDNA3.1-L (1 µg) using 15µL of Lipofectamine 2000 transfection reagent (InvitrogenTM). The plasmid mixture was mixed in 1 mL reduced-serum medium Gibco™ Opti-MEM™ containing 1 focus-forming unit per cell of modified vaccinia virus expressing T7 RNA polymerase (MVA-T7). The whole plasmid mixture in Opti-MEM medium was used to transfect HEp2 cells in a six-well plate for six hours. After transfection, the cells were washed twice and incubated in DMEM containing 10% allantoic fluid and 2% FBS. Three days later, the whole cell culture was frozen until being injected into 10-day-old SPF ECEs for virus recovery. Eggs allantoic fluid was collected three days post-injection and the virus recovery was determined by HA assay. Positive samples were further propagated in 10-day-old SPF ECE and the genome of the recovered virus was sequenced in its entirety. Presence of the newly introduced and deleted RE sites from the viral genome were used as genetic markers to confirm the recovery of the rAPMV3-Wisc. virus. 

### 2.5. Construction and Recovery of Recombinant APMV-3 (rAPMV-3) Strain Wisc. Expressing Green Fluorescent Protein (GFP)

The cDNA of enhanced GFP gene was inserted at the PmeI site in the P-M gene junction of the pAPMV-3 Wisc. FLC. The GFP ORF was flanked by the M gene-start and P gene-end sequences of APMV-3 Wisc. A Kozak sequence was inserted before the GFP ORF for enhanced translation. The length of the inserted gene cassette (822 nucleotides) was adjusted to a multiple of six by adding four nucleotides after the GFP ORF following the rule of six [25,26] (Figure 2A). Recombinant APMV-3 Wisc. expressing GFP (rAPMV-3 Wisc.\GFP) was recovered using the same procedure mentioned above. GFP expression by the recovered virus was observed in DF-1 cells (Figure 2B). In order to ensure consistent and efficient expression of GFP by the recombinant virus, the recovered virus was plaque purified twice in DF-1 cells and passed in eggs for eight serial passages before being tested again for the presence of GFP gene by RT-PCR and GFP expression in DF-1 cells. APMV-3 Wisc. did not produce visible plaques in DF-1 cells under methyl cellulose overlay like APMV-3 Neth. Hence, rAPMV-3 Wisc.\GFP was purified by infecting DF-1 cells at high dilutions and covering it with 0.8% methyl cellulose overlay medium containing 10% allantoic fluid and 2% FBS. Two days post-infection, single fluorescent foci of rAPMV-3 Wisc.\GFP were picked and propagated in 10-day-old SPF ECE.

### 2.6. Multicycle Growth Kinetics of the Constructed Recombinant Viruses and Wild-Type APMV-3 Wisc

Multicycle growth kinetics of wild type APMV-3 Wisc., rAPMV-3 Wisc. and rAPMV-3 Wisc.\GFP were determined in DF-1 cells. Eighty percent confluent DF-1 cells in six well plate were infected by 0.01 multiplicity of infection (MOI) of each virus then 200µL cell supernatants were collected at 12-h intervals for three days. The viruses used for infection and the collected cell supernatants were titrated in DF-1 cells by immunostaining to count the virus fluorescent foci [27]. The focal fluorescent unit count (FFU/mL) was obtained by infecting DF-1 cells in 24-well plates with 10-fold serially diluted virus and covering it with 0.8% methylcellulose overlay medium containing 10% allantoic fluid and 2% FBS. Two wells were infected for every dilution and the average count was calculated. Two days post-infection, the overlay was removed and the cells were fixed and permeabilized by methanol for 30 minutes. Fixed cells were washed twice by phosphate-buffered saline (PBS, 5 minutes each) followed by blocking using 3% goat serum for 30 minutes. Immunostaining was done using rabbit anti-APMV-3 N protein primary antibody (0.5%) for two hours. The cells were then washed four times by PBS and incubated for 1 h with Alexa flour labelled goat anti-rabbit secondary antibody (0.1%). Cells were then washed three times in PBS, the virus fluorescent foci were counted under a fluorescent microscope and the virus titer was calculated.

### 2.7. Comparison of rAPMV-3 Wisc., rAPMV-3 Netherlands and Strain LaSota (rLasota) Vectors Expressing GFP In Vitro

The three recombinant avian paramyxovirus vectors expressing GFP (rAPMV-3 Wisc.\GFP, rAPMV-3 Neth.\GFP and rLasota\GFP) were compared for their growth rate and GFP expression (as a foreign protein) in DF-1 cells. Multicycle growth kinetics of the three vectors was compared in DF-1 cells using the aforementioned protocol for growth kinetics. The collected virus aliquots were titrated in DF-1 cells using the above mentioned protocol to count the virus fluorescent foci per ml without immunostaining (GFP fluorescence was used instead) (Figure 3A). 

GFP expression by the three recombinant viruses was measured in DF-1 cells using Western blot. DF-1 cells in a twelve-well plates were infected by 0.5 MOI of each virus. Cell lysate from each well was collected at 24 and 48 h post-infection using 120 µL Radioimmunoprecipitation assay (RIPA) lysis buffer. The collected lysate was kept on ice for 15 mins then centrifuged at 15,000× g for 15 mins. The Lysate supernatant was separated and mixed with 6x protein loading dye, boiled for 10 mins and subjected to 12% sodium dodecyl sulfate polyacrylamide gel electrophoresis (SDS-PAGE). GFP amount was measured by Western blot using a rabbit anti-GFP and normalized against the cellular protein B-tubulin measured by mouse anti B-tubulin serum as a loading control.

### 2.8. Evaluation of rAPMV-3 Wisc.\GFP, rAPMV-3 Neth.\GFP and rLasota\GFP as Vaccine Vectors in Chickens

Four groups of day-old SPF chickens (nine chickens per group) were housed in negative pressure isolators in biosafety level 2 animal facility. Feed and water were provided ad libitum. Three groups of 1-day-old chickens were intraocular vaccinated by 10^6^ FFU of each virus vector per bird using fresh allantoic fluid. The remaining group was mock infected by PBS as a negative control (Appendix A). Back titration of the viruses used in vaccination was 1–3 (10^5^) FFU of each virus per bird. Three days post-infection, three chickens were euthanized from each group and different organs were collected for virus detection and titration. Half of the brain, both lungs, trachea and spleen of each chicken were separately homogenized in 1.5 mL DMEM containing 5x antibiotic (Pen Strep) (only the spleen was homogenized in 1 mL DMEM). The organ homogenates were clarified and the supernatants were titrated for the vaccinating viruses in DF-1 cells. Duplicate wells in 24-well plates were infected by 250 µL of (2–1, 10–1, 10–2) dilutions of the organ homogenates and virus titers were calculated as focal fluorescent unit per organ. The remaining chickens were observed daily for any clinical signs of illness for 10 days and weighed two weeks post-infection. Two weeks post immunization; serum samples were collected from the six chickens in each group to evaluate the induced antibody immune response against the used vector by haemagglutination inhibition (HI) assay. HI assay was performed using 4 haemagglutinating (HA) units of each vaccinating virus vector. The animal experiment was undertaken following guidelines and after approval of the Animal Care and Use Committee (IACUC) and Institutional Biosecurity Committee (IBC), University of Maryland. 

## 3. Results

### 3.1. Recovery of rAPMV-3 Wisc. by a Reverse Genetics System

Antigenomic cDNA fragments of APMV3-Wisc. were synthesized by RT-PCR from genomic RNA and cloned into the pBR322/dr using the designed polylinker. Sequences of the prepared cDNA fragments were confirmed to ensure the introduced and deleted RE sites (Appendix A) in the genome without any unintended mutations from the wild-type APMV-3 Wisc. sequence (GenBank accession number, EU782025). Three G residues were included at the 5′ end of T7 promotor to improve the transcription efficiency. APMV-3 Wisc. L-gene was cloned into three different expression plasmids; pTM, pCDNA3.1 and pGEM7zf, as trials to recover the virus, however none of them was able to recover the virus. Hence, APMV-3 Neth. pCDNA3.1-L was used instead, with APMV-3 Wisc. pTM-N and -P support plasmids to recover the rAPMV-3 Wisc. in HEp-2 cells. Virus recovery was confirmed by injecting ECE with the collected cell lysate and testing its allantoic fluid for HA activity. The recovered virus was serially passed three times in 10-day-old SPF ECE to remove the co-infected vaccinia virus. The virus genomic RNA was isolated from positive allantoic fluid and confirmed to have the introduced genetic markers of the rAPMV-3 Wisc. construct by complete genome sequencing.

### 3.2. Construction and Recovery of rAPMV-3 Wisconsin Expressing GFP

Enhanced GFP gene was cloned at the PmeI site in the P-M gene junction of rAPMV-3 Wisc. The inserted GFP cassette was confirmed by sequencing. The rAPMV-3 Wisc.\GFP was recovered using the described protocol and used to infect DF-1 cells. GFP expression was visualized under a fluorescent microscope (Figure 2B). The virus was passaged twice in SPF ECE, plaque purified in DF-1 cells and passed for eight serial passages in SPF ECE to confirm stability of GFP expression. GFP gene stability was confirmed by RT-PCR and GFP expression in DF-1 cells until the last passage (data not shown). Multicycle growth kinetics of wild-type APMV-3 Wisc., rAPMV-3 Wisc. and rAPMV-3 Wisc.\GFP in DF-1 cells were plotted as shown in (Figure 2C). Each result represents the mean titer of two different experiments. During the first two days, rAPMV-3 Wisc.\GFP showed an average of one log_10_ less titer than that of rAPMV-3 Wisc. and about two log_10_ less titer than that of wild-type APMV-3 Wisc. By the end of the third day, the three viruses reached similar titers as all the available cells got infected (plateau phase). 

### 3.3. Multicycle Growth Kinetics and GFP Expression of rAPMV-3 Wisc.\GFP, rAPMV-3 Neth.\GFP and rLasota\GFP In Vitro

Multicycle growth kinetics of the three APMV vectors expressing GFP showed that rAPMV-3 Neth.\GFP had the highest growth rate reaching about 10-fold the virus titer of rLasota\GFP and 100-fold the virus titer of rAPMV-3 Wisc.\GFP at 36 h post-infection (Figure 3B). The titers plotted in the graph represent the mean of two different growth kinetics experiments. GFP expression in DF-1 infected cells was compared among the three virus vectors at 24 and 48 h post-infection using Western blot (Figure 3C). One-day after infection, rAPMV-3 Wisc.\GFP showed the least expression of GFP followed by rLasota\GFP (about two times that of rAPMV-3 Wisc.\GFP), while rAPMV-3 Neth.\GFP showed the highest expression (about three times that of rAPMV-3 Wisc.\GFP). Two days post-infection, rAPMV-3 Wisc.\GFP and rLaSota\GFP showed similar levels of GFP expression, while rAPMV-3 Neth.\GFP remained to be the highest expressing virus vector (1.5X that of rAPMV-3 Wisc.\GFP). Cellular B-tubulin was used to normalize the amount of the expressed GFP to the amount of the loaded cell lysate per well.

### 3.4. Growth Characteristics and Antibody Response of rAPMV-3 Wisc.\GFP, rAPMV-3 Neth.\GFP and rLasota\GFP in Chickens

One-day-old SPF chickens in groups of nine were inoculated by 1–3 × 10^5^ FFU of each virus per bird (by the ocular route). Brain, lung, trachea and spleen were collected separately from three birds of each group, three days’ post-infection, homogenized and titrated for the corresponding viruses. rLaSota\GFP showed the highest mean virus titer in the trachea (104–105 FFU/organ) with no evidence of the virus in the brain or spleen (Figure 4A). Whereas, rAPMV-3 Neth.\GFP showed high virus titers in all examined organs; trachea, brain, spleen and lungs (with mean virus titer of 103–104 FFU/organ). rAPMV-3 Wisc.\GFP was detected only in the trachea with a titer of (102 FFU/organ) (Figure 4A). 

The remaining chickens in each group were observed daily for signs of illness for 10 days and body weighed two weeks post-infection. No clinical signs were observed in all groups except for chickens infected by the rAPMV-3 Neth.\GFP. They showed early decrease in food consumption, stunted growth and abnormal feathering after vaccination (Appendix A). Their average body weights was 20% lower than that of the control group. On the other hand, both rLaSota\GFP and rAPMV-3 Wisc.\GFP groups did not show a significant change in body weights compared to the control group (Figure 4C). The serum samples of chickens collected 14 days’ post-infection have shown that all the three vaccinated chicken groups were seroconverted when compared to the control group. The mean HI titer of the chickens infected with rLaSota\GFP, rAPMV-3 Wisc.\GFP, and rAPMV-3 Neth.\GFP were 2^3.5^, 2^6^, 2^7^ respectively (Figure 4B).

## 4. Discussion

Avian paramyxoviruses have been employed as vaccine vectors against different avian and human pathogens for two decades [4,5,11]. However, APMV-1 vector was reported to be interfered with by the maternally derived antibodies (MDA) against NDV in young commercial chickens, leading to vaccination failure [28]. To overcome this obstacle, chimeric APMV-1 [29,30] or other antigenically distinct APMVs [9,20] are used as vaccine vectors to escape neutralization by MDA of APMV-1. Among other APMVs, APMV-3 has proven to be a promising vector for poultry vaccination [7,20]. 

To date, two strains of APMV-3 have been fully characterized: APMV-3 strain Wisconsin and APMV-3 strain Netherlands. Both strains are considered members of the same serotype, but their complete genome sequence, reciprocal HI and virus neutralization have revealed that they represent two antigenic subgroups of APMV-3 [17]. Although both APMV-3 strains are considered avirulent [16], APMV-3 strain Netherlands was shown to be mildly pathogenic for day-old chickens, whereas APMV-3 strain Wisconsin was completely non-pathogenic for day-old chickens [16,23]. In addition, APMV-3 strain Wisconsin does not kill ECE upon injection; whereas, APMV-3 strain Netherlands kills ECE [16]. For these reasons, we expected APMV-3 strain Wisconsin to be a safer vaccine vector for young chickens. Although a reverse genetics system was created and evaluated for APMV-3 strain Netherlands [7,21], there was no reverse genetics system available for APMV-3 strain Wisconsin. 

In this study, we created a plasmid-based reverse genetics system for APMV-3 strain Wisconsin and used the recombinant virus to express GFP as a foreign protein in vitro and in vivo. For APMV-3 strain Wisconsin recovery, the L gene of APMV-3 strain Netherlands was used as a support plasmid instead of that of APMV-3 strain Wisconsin as our initial trials to recover the Wisconsin strain using its own L gene failed. We used the L gene of APMV-3 strain Netherlands as both strains belong to the same serotype and strain Netherlands replicates faster than the strain Wisconsin [16]; therefore, it may have a robust polymerase (L) protein. APMV-3 strain Wisconsin was successfully recovered using the heterologous L gene. Complete genome sequencing of the recovered virus confirmed the absence of any recombination and the presence of our introduced mutations (Appendix A). We also noticed that APMV-3 strain Wisconsin can be recovered more efficiently using all three support plasmids (N, P, and L) of APMV-3 strain Netherlands rather than a mix of the strain Wisconsin N and P plasmids and the strain Netherlands L plasmid. This finding suggests the lower polymerase activity of APMV-3 strain Wisconsin L protein compared to that of APMV-3 strain Netherlands and ensures the vital role played by the L protein for APMV recovery. It also indicate that the polymerase complex proteins (N, P, and L) of APMV-3 strain Netherlands are capable of recognizing the cis-acting regulatory sequences (leader and trailer) of APMV-3 strain Wisconsin and support its replication. Similar findings were shown in previous reports where the polymerase complex proteins of one virus can support replication of a closely related virus belonging to the same genus [31,32,33]. However, polymerase protein genes of APMV-1 did not support the replication of APMV-3 strain Wisconsin (our unpublished results).

The development of a reverse genetics system for paramyxoviruses is a challenging task and some APMVs cannot be recovered till now [10]. Our study confirms this observation, as APMV-3 strain Wisconsin was non-recoverable using its own L gene support plasmid. Hence, the polymerase protein weakness of some APMVs may be the reason of their unsuccessful recovery. It is also worth mentioning that the L gene was found to be the second major determinant of APMV-1 pathogenicity after the F gene [34,35]. Therefore, it would be interesting to determine whether replacing the L gene of APMV-3 strain Wisconsin with that of APMV-3 strain Netherlands would increase the growth rate and pathogenicity of the chimeric Wisconsin strain. 

Multicycle growth kinetics showed one log_10_ lower virus titer of the recovered APMV-3 Wisconsin than its parental wild-type virus. Similar growth retardation in other recombinant APMVs was observed when compared to their parental virus growth rate [9]. In our recombinant virus, the reason for its delayed growth rate could be the genetic mutations (18 changed nucleotides) that were introduced in its genomic untranslated regions (Appendix A). Another possible explanation is that the wild-type APMV-3 strain Wisconsin is composed of quasispecies (collection of variants), whereas the recombinant virus represents an individual molecular clone and has undergone far less variation.

The recovered virus was successfully used as a vector to express GFP as a foreign protein and confirmed the stable insert expression for at least 10 consecutive passages in ECE. The growth rate of the rAPMV-3 Wisconsin with GFP was about 1 log_10_ lower than that of the parental recombinant virus and 2 log_10_ lower than that of the wild-type virus for the first 48 h post-infection. Similar growth retardation was observed in other paramyxoviruses bearing foreign genes [5,36,37]. Although rAPMV-3 Wisconsin successfully expressed the GFP gene (700 nucleotides), its ability to express foreign genes of larger size is unknown. 

The growth kinetics and GFP expression of rAPMV-3 Wisc.\GFP was compared against rAPMV-3 Neth.\GFP and rLaSota\GFP in DF-1 cells. rAPMV-3 Neth.\GFP showed the highest growth rate reaching about 10-fold the rLasota\GFP titer and 100-fold the rAPMV-3 Wisc.\GFP titer at 36 h post-infection. In agreement with growth kinetics, GFP expression was the highest in rAPMV-3 Neth.\GFP, followed by rLaSota\GFP and rAPMV-3 Wisc.\GFP. This finding was supported by the difference in size of their fluorescent foci at 48 h post-infection (Figure 3A). The remarkable difference observed between the growth patterns of the two APMV-3 strains may be due to the difference in their polymerase complex activity and their F proteins cleavage site sequence.

An in vivo study was performed to compare tissue distribution and humoral antibody response of the three APMV vectors in chickens. Our results showed that rAPMV-3 strain Netherlands propagated in different body organs; brain, lung, trachea, and spleen with remarkable titers after intraocular vaccination of day-old chickens, whereas rAPMV-3 strain Wisconsin and rLaSota vectors were confined only to the respiratory tract by the third day post-infection. The wide propagation of APMV-3 strain Netherlands came at the expense of mild pathogenicity observed in young chickens shortly after their vaccination. APMV-3 strain Netherlands vaccinated group showed decreased feed intake, early retarded growth (Appendix A), and abnormal (mal) feathering. A similar observation of stunted growth was previously reported in young chickens experimentally infected by APMV-3 strain Netherlands [23]. On the other hand, both rAPMV-3 strain Wisconsin and rLaSota vaccinated groups did not show any observable pathogenicity. 

Recombinant LaSota virus replication was the highest in the respiratory tract among the three virus strains used in this study. Hence, the rLaSota may be a suitable vaccine vector for protection against poultry respiratory viruses that require strong mucosal immunity, e.g. infectious laryngotrachitis virus. The wide tissue tropism of APMV-3 strain Netherlands could be due to the multi-basic amino acid sequence in its F-protein cleavage site (FPCS) (Appendix A). This allows it to be cleaved by the ubiquitous intracellular furin protease available in different tissue organs; whereas APMV-3 strain Wisconsin and APMV-1 strain LaSota are restricted to the respiratory tract as their FPCS has monobasic amino acid which is cleavable only by extracellular trypsin-like protease available in certain tissue types (e.g., respiratory tract) [9,19,38]. As rAPMV-3 strain Netherlands replicates systemically in different body organs, it induced three-fold higher HI titer than that induced by APMV-1 strain LaSota. Therefore rAPMV-3 strain Netherlands may be a better vector for protection against pathogens requiring a strong systemic immune response, e.g., avian influenza or infectious bursal disease virus. Recent studies have also shown that APMV-3 strain Netherlands vector induced a higher immune response against the expressed foreign antigen (Ebola virus glycoprotein or avian influenza HA) than that induced by rLaSota vector [20,22]. 

## 5. Conclusions

In conclusion, in this study we developed a reverse genetics system for APMV-3 strain Wisconsin to be used as a safe vaccine vector for one day-old chickens. We have also compared the biological properties of APMV-1 strain LaSota, APMV-3 strain Netherlands and APMV-3 strain Wisconsin as viral vectors in vitro and in vivo.

## Figures and Tables

**Figure 1 viruses-13-00316-f001:**
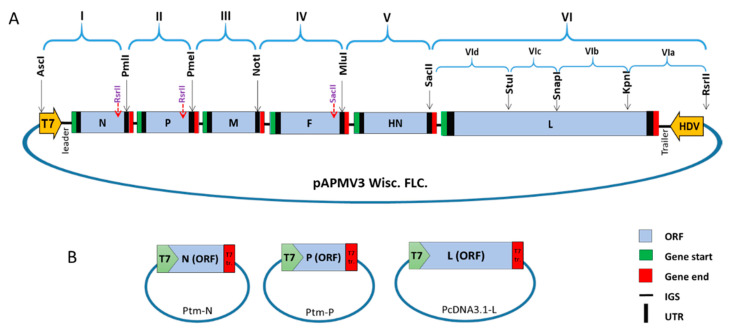
Construction of avian paramyxovirus 3 strain Wisconsin full-length expression plasmid (pAPMV-3 Wisc.) full-length clone (FLC) cDNA and its support plasmids (**A**) The full-length cDNA clone was constructed by assembling six sub-genomic fragments into pBR 322/dr using a 116 nucleotide long oligonucleotide linker to form an antigenomic full-length cDNA clone (pAPMV-3 Wisc.). (**B**) Three support plasmids were constructed by individually cloning the N, P, and L genes into T7 polymerase expression plasmids (pTM for the N and P genes and pCDNA 3.1 for the L gene).

**Figure 2 viruses-13-00316-f002:**
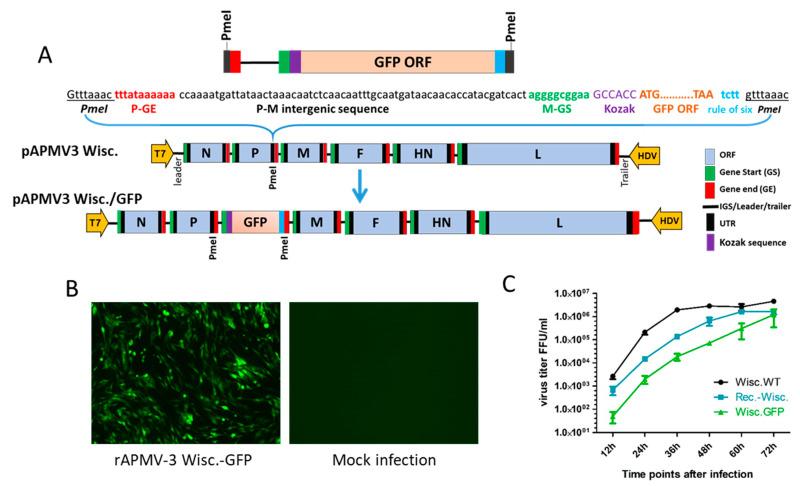
Construction of recombinant APMV-3 Wisc. expressing green fluorescent protein (GFP). (**A**) GFP gene cassette was constructed by including P gene-end, P-M intergenic sequence, M gene-start, and Kozak sequence for enhanced translation. The GFP cassette was cloned between P and M genes at the PmeI site. The full length cDNA clone was termed pAPMV3 Wisc.\GFP. (**B**) GFP expression by recombinant APMV-3 Wisc.\GFP (rAPMV-3 Wisc.\GFP) in DF-1 cells two days post-infection. (**C**) Multicycle growth kinetics of wild type APMV-3 Wisc., rAPMV-3 Wisc., and rAPMV-3 Wisc.\GFP in DF-1 cells using 0.01 multiplicity of infection for each virus. The virus titers were calculated as a mean of two different growth kinetics experiments in FFU/mL.

**Figure 3 viruses-13-00316-f003:**
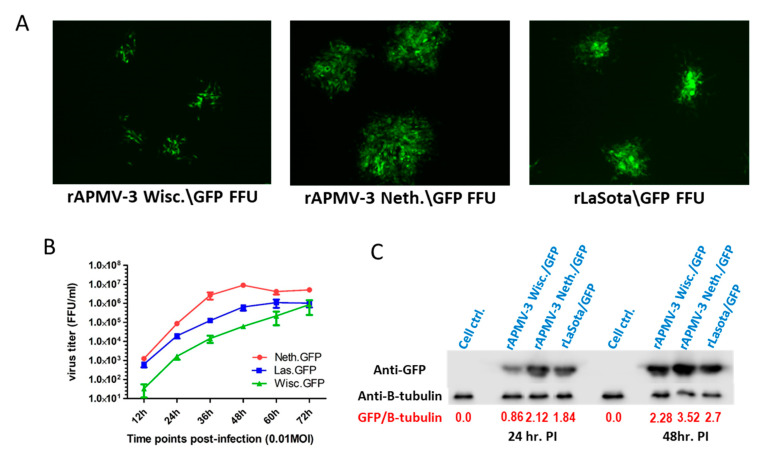
Characterization of rLaSota\GFP, rAPMV-3 Wisc.\GFP and rAPMV-3 Neth.\GFP in DF-1 cells. (**A**) Fluorescent foci of the three recombinant vectors expressing GFP in DF-1 cells two days-post-infection. (**B**) Multicycle growth kinetics of the three viruses in DF-1 cells at 12 h intervals using 0.01 MOI of each virus. (**C**) Western blot for GFP expressed by the three viral vectors using 0.5 MOI of each virus in DF-1 cells. Cell lysate was collected 24 and 48 h post-infection and analyzed by Western blot using rabbit GFP antibodies and mouse B-tubulin antibodies.

**Figure 4 viruses-13-00316-f004:**
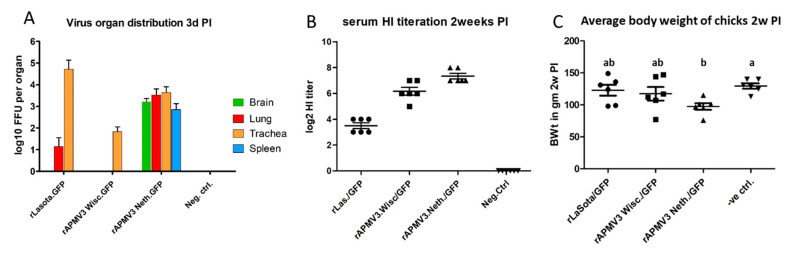
Replication of rLaSota\GFP, rAPMV-3 Wisc.\GFP and rAPMV-3 Neth.\GFP in SPF chickens. (**A**) Three birds of each immunized group were sacrificed 3 days after infection and different organs (brain, lung, trachea, and spleen) from each bird were titrated for the used APMV vector in DF-1 cells. The plotted virus titers in different organs are expressed as the mean FFU per organ. (**B**) Serum samples of chickens immunized by rLaSota\GFP, rAPMV-3 Wisc.\GFP and rAPMV-3 Neth.\GFP were analyzed two weeks after infection for HI activity against 4 HAU of the vaccinating virus. Results represent the mean log_2_ HI titers of six birds from each immunized group. (the error bar represents standard error of the mean (SEM)). (**C**) Average body weight of chicken groups (gm) two weeks post vaccination.

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
