# Peer review of "Recovery of Recombinant Avian Paramyxovirus Type-3 Strain Wisconsin by Reverse Genetics and Its Evaluation as a Vaccine Vector for Chickens"

_viruses, 2021, doi:10.3390/v13020316_

Round 1

Reviewer 1 Report

This is an interesting paper describing efforts to construct by revertive genetics system avian paramyxovirus type 3 recombinant vaccine. This approach takes into account animal trial on 1-d old chickens.  The created recombinant vaccine seems to be an alternative for currently applied commercial vaccines. Although, the paper meets the interest on current avian paramyxoviruses prevention and prophylaxis i requires some language corrections. Ex. 'chicks' should be rephrased to ' chickens'. Therefore, the revision of this paper by native English-speaking person seems to by compulsory.

Author Response

Thank You for your comments.
As suggested by you we have made language corrections and rephrased the word "chicks" to "chickens" all over the manuscript including the title.

Reviewer 2 Report

The authors developed a reverse genetics system for APMV-3 strain Wisconsin. This strain is less pathogenic for chickens than APMV-3 strain Netherlands. By generating a recombinant virus that expresses the GFP gene, the authors show that APMV-3 Wisconsin can be used as a vector for the expression of a foreign protein. The fact that APMV-3 Wisconsin is safe for 1-day-old chickens and is not affected by pre-existing immunity against APMV-1 (Newcastle disease virus) makes it an attractive alternative for APMV-1 based vector vaccines.

Overall, this is a well-designed study, and the experimental approach is sound and solid. Furthermore, the paper is well written. Nevertheless, I do have a few questions that the authors should address.

The authors’ claim that the APMV-3 Wisconsin vector is a suitable alternative for APMV-1 LaSota is not fully addressed in this paper since they only used reporter-gene expression and did not measure the immune-response against a relevant immunogen. Although the data indicate that Wisc.GFP(P8) replicates a little faster than Las.GFP (Fig. 3B) and the HI-titre induced by Wisc.GFP(P8) is higher than that induced by Las.GFP (Fig. 4B), expression of the foreign gene (GFP) is higher in Las.GFP infected cells compared to Wisc.GFP(P8) infected cells (Fig. 3C). Since antigen mass is an important factor that determines the outcome of vaccination the question remains whether Wisconsin strain is indeed a better vaccine vector than strain LaSota.

Related to this, how do the authors explain the higher expression level of GFP by LaSota compared to the Wisconsin strain?

According to the new ICTV classification, the genus Avian paramyxovirus (APMV) has been renamed Avian orthoavulavirus.

The difference in growth-kinetics in DF-1 cells between the wild-type Wisc.WT and Rec.-Wisc is significant (Fig. 2C). The authors note that this may be due to nucleotide differences that were introduced in order to assemble the full-length genome from several subgenomic restriction fragments (Fig. 1A). Since all nucleotide differences are either located in untranslated regions or are missense mutations, I wonder whether this is the real cause. An alternative explanation would be that the Wisc.WT virus consists of a quasispecies (a collection of variants) whereas the Rec.-Wisc virus is derived from an individual molecular clone and has undergone far less variation.

The difference in growth-kinetics between Rec.-Wisc and Wisc.GFP is also quite remarkable (Fig. 2C). Others have shown that the insertion of the GFP gene or another foreign gene between the NP and P genes hardly affected the growth kinetics of the resulting viruses (e.g. Zang et al., J. Gen. Virol. (2015); Pan et al. (2016) PLoS ONE 11(10) 96, 2028–2035; Chelappa et al., 2017, Virus Genes 53:410–417). Could these differences be related to the design of the vector?

The authors note that an expression plasmid containing the L-gene from APMV-3 Wisc could not be used to rescue the Wisconsin virus, whereas a plasmid containing the L-gene from APMV-3 Neth could. They argue that this may be related to the activity of the L-protein.
In this respect it is worthwhile to note that rescue of APMV-1 LaSota using the LaSota L-protein that - judging from the replication kinetics shown in Fig. 3B – has an activity that is comparable or even lower than that of APMV-3 Wisconsin was successful.
Did the authors try to rescue LaSota using the Wisconsin L-gene? Or rescue Wisconsin using the LaSota L-gene? (same genus)

The lack of superscript to indicate the power of 10 or 2 is very confusing. I was puzzled for some time by the HI-values given in line 330 and the data shown in Fig. 4B. (23.5 instead of 2^3.5, etc.).

Author Response

We thank the reviewer for his comments.

Attached is our itemized answer for all his comments.

We thank the reviewer for the comments. We have revised the manuscript based on reviewer’s comments. Our answer to the reviewer comments are given below;

Point 1: The authors’ claim that the APMV-3 Wisconsin vector is a suitable alternative for APMV-1 LaSota is not fully addressed in this paper since they only used reporter-gene expression and did not measure the immune-response against a relevant immunogen.

Response 1: we agree with the reviewer that whether APMV-3 Wisconsin vector is a suitable alternative to APMV-1 LaSota vector is not fully addressed in this paper. It would require rigorous comparison using an immunogen of actual poultry pathogen rather than GFP. We think it can act as an alternative vaccine vector for LaSota as the recombinant APMV-3 strain Wisconsin expressing GFP successfully replicated in chickens’ respiratory tract and induced humoral immune response as measured by HI assay.

Point 2: Although the data indicate that Wisc.GFP(P8) replicates a little faster than Las.GFP (Fig. 3B) and the HI-titer induced by Wisc.GFP(P8) is higher than that induced by Las.GFP (Fig. 4B), expression of the foreign gene (GFP) is higher in Las.GFP infected cells compared to Wisc.GFP(P8) infected cells (Fig. 3C). Since antigen mass is an important factor that determines the outcome of vaccination the question remains whether Wisconsin strain is indeed a better vaccine vector than strain LaSota. Related to this, how do the authors explain the higher expression level of GFP by LaSota compared to the Wisconsin strain?

Response 2: rLasota\GFP replicates at higher rate than rAPMV-3 Wisc.\GFP. Our fore mentioned growth curve in fig.3 compared the growth rate of the eighth passage of rAPMV-3 Wisc.\GFP (P8), which became a little faster than rAPMV-3 Wisc.\GFP earlier passages. If you compared the rAPMV-3 Wisc.\GFP growth curve titers used in fig.2, you will notice its titers are lower than those recorded in fig.3 for rAPMV-3 Wisc.\GFP (P8). Since the earlier passages of rAPMV-3 Wisc.\GFP were used for GFP expression experiments (Fig.3) and animal experiment (fig.4). We replaced the growth curve in fig3 with another one was done using the earlier passage of rAPMV-3 Wisc.\GFP, rAPMV-3 Neth.\GFP and rLasota\GFP. The new growth curve figure was done using two replicates of each virus vector using the same conditions and MOI (0.01).

Point 3: According to the new ICTV classification, the genus Avian paramyxovirus (APMV) has been renamed Avian orthoavulavirus.

Response 3: as suggested by the reviewer, we have mentioned in our revised manuscript the new classification and nomenclature of APMVs mentioned by the ICTV (page.2, lines.33-37)

Point 4: The difference in growth-kinetics in DF-1 cells between the wild-type Wisc.WT and Rec.-Wisc is significant (Fig. 2C). The authors note that this may be due to nucleotide differences that were introduced in order to assemble the full-length genome from several subgenomic restriction fragments (Fig. 1A). Since all nucleotide differences are either located in untranslated regions or are missense mutations, I wonder whether this is the real cause. An alternative explanation would be that the Wisc.WT virus consists of a quasispecies (a collection of variants) whereas the Rec.-Wisc virus is derived from an individual molecular clone and has undergone far less variation.

Response 4: we agree with the reviewer explanation to the difference in growth kinetics between the wild type wisc. and the recombinant Wisc. could be due to the quasispecies present in the wild type virus and we added that to our discussion, page.12 line 329-331.

Point 5: The difference in growth-kinetics between Rec.-Wisc and Wisc.GFP is also quite remarkable (Fig. 2C). Others have shown that the insertion of the GFP gene or another foreign gene between the NP and P genes hardly affected the growth kinetics of the resulting viruses (e.g. Zang et al., J. Gen. Virol. (2015); Pan et al. (2016) PLoS ONE 11(10) 96, 2028–2035; Chelappa et al., 2017, Virus Genes 53:410–417). Could these differences be related to the design of the vector?

Response 5: We agree with the reviewer that the difference in growth-kinetics between recombinant Wisc. and Wisc.GFP could be related to the vector design. In general, insertion of a foreign gene between P and M genes has minimal effect on growth kinetics of the resulting virus. But this is not the case in all genes. some genes may be better to insert between N and P. we will keep that point in mind in our future studies.

Point 6: The authors note that an expression plasmid containing the L-gene from APMV-3 Wisc could not be used to rescue the Wisconsin virus, whereas a plasmid containing the L-gene from APMV-3 Neth could. They argue that this may be related to the activity of the L-protein.
In this respect it is worthwhile to note that rescue of APMV-1 LaSota using the LaSota L-protein that - judging from the replication kinetics shown in Fig. 3B – has an activity that is comparable or even lower than that of APMV-3 Wisconsin was successful.
Did the authors try to rescue LaSota using the Wisconsin L-gene? Or rescue Wisconsin using the LaSota L-gene? (same genus)

Response 6: We don’t have any problems in recovery of APMV-1 LaSota strain using its own L-gene. However, we tried to rescue APMV-3 strain Wisconsin by LaSota L-gene and it did not work. Mostly because they belong to two different genera; Orthoavulavirus (APMV-1) and Metaavulavirus (APMV-3), and have different polymerase recognition sequences in their leader and trailer sequnces. We have added that point to our revised manuscript (page 12, lines 316-317)

Point.7: The lack of superscript to indicate the power of 10 or 2 is very confusing. I was puzzled for some time by the HI-values given in line 330 and the data shown in Fig. 4B. (23.5 instead of 2^3.5, etc.).

Response 7: we apologize for the unintended typing mistakes and have corrected them in our revised manuscript.
